# Donor Lymphocyte Infusion in the Treatment of Post-Transplant Relapse of Acute Myeloid Leukemias and Myelodysplastic Syndromes Significantly Improves Overall Survival: A French–Italian Experience of 134 Patients

**DOI:** 10.3390/cancers16071278

**Published:** 2024-03-26

**Authors:** Eugenia Accorsi Buttini, Cristina Doran, Michele Malagola, Vera Radici, Marco Galli, Vicky Rubini, Alessandro Leoni, Mirko Farina, Nicola Polverelli, Federica Re, Simona Bernardi, Mohamad Mohty, Domenico Russo, Eolia Brissot

**Affiliations:** 1Unit of Blood Diseases and Bone Marrow Transplantation, Cell Therapies and Hematology, Research Program, Department of Clinical and Experimental Science, University of Brescia, ASST Spedali Civili di Brescia, 25123 Brescia, Italy; michele.malagola@unibs.it (M.M.); vera.radici@unibs.it (V.R.); m.galli020@studenti.unibs.it (M.G.); vicky.rubini@unimi.it (V.R.); mirko.farina@unibs.it (M.F.); simona.bernardi@unibs.it (S.B.); domenico.russo@unibs.it (D.R.); 2Service d’ Hématologie Clinique et Thérapie Cellulaire, Hôpital Saint Antoine, Centre de Recherche Saint-Antoine (CRSA), Sorbonne Université, 75012 Paris, Franceeolia.brissot@aphp.fr (E.B.); 3Research Center Ail (CREA), Department of Clinical and Experimental Science, University of Brescia, ASST Spedali Civili di Brescia, 25123 Brescia, Italy; alessandro.leoni@unibs.it (A.L.); federica.re@unibs.it (F.R.); 4Division of Hematology, Fondazione IRCCS Policlinico San Matteo, 27100 Pavia, Italy; n.polverelli@smatteo.pv.it

**Keywords:** allogeneic stem cell transplantation, acute myeloid leukemia, myelodysplastic syndrome, relapse, donor lymphocyte infusion, relapse treatment, graft-versus-leukemia

## Abstract

**Simple Summary:**

Allogeneic stem cell transplantation (allo-SCT) represents the only potentially curative treatment for high-risk acute myeloid leukemia (AML) and myelodysplastic syndrome (MDS), but up to 50% of patients relapse after allo-SCT. The salvage therapy after disease recurrence is not standardized, and the outcome remains unfavorable. Therefore, there is a growing interest in determining the most effective approach to manage the post-transplant phase with the goal of promptly detecting disease recurrence or preventing it. In this context, we conducted a retrospective study to assess the overall survival (OS) of patients with relapsed AML or MDS after allo-SCT with the aim of acquiring useful information for identifying the best prospective therapeutic strategy. The OS was evaluated according to the type of therapy, whether it included donor lymphocyte infusion (DLI), the timing of administration, and whether it occurred during an overt hematological relapse or in a preemptive setting.

**Abstract:**

Background: Disease relapse after allogeneic stem cell transplantation (allo-SCT) is the main challenge for curing acute myeloid leukemia (AML) and myelodysplastic syndrome (MDS). We investigated the overall survival (OS) after allo-SCT relapse according to different therapeutic approaches. Methods: We analyzed 134 patients who relapsed after allo-SCT performed between 2015 and 2021 at Saint-Antoine University Hospital, Paris and Spedali Civili di Brescia, Brescia. Of these, 103 (77%) were treated, comprising 69/103 (67%) who received therapy in overt relapse and 34/103 (33%) who were treated in a pre-emptive manner when molecular/cytogenetics recurrence or mixed chimerism occurred. The treatment was donor lymphocyte infusion (DLI)-based for 40/103 (39%) patients. Results: The 1-, 2-, and 5-year OS of patients treated with DLI *(n* = 40) was 67%, 34%, and 34%, respectively, for those treated preventively (n = 20) and 43%, 20%, and 20%, respectively, for those treated in overt relapse (n = 20) (*p* < 0.01). The 1-, 2-, and 5-year OS of patients treated without DLI (n = 63) was 54%, 40%, and 26%, respectively, for those treated preventively (n = 14) and 17%, 5%, and 0%, respectively, for those treated in overt relapse (n = 49) (*p* < 0.01). Conclusions: Relapse treatment with a pre-emptive strategy was associated with improved outcomes, particularly when DLI was employed.

## 1. Introduction

Allogeneic hematopoietic stem cell transplantation (allo-SCT) is recognized as the most potent anti-leukemic treatment for patients with acute myeloid leukemia (AML) and myelodysplastic syndrome (MDS) [1,2]. In recent years, the introduction of reduced-intensity conditioning (RIC) regimens, advancements in graft-versus-host disease (GVHD) prophylaxis, and improvements in supportive care have reduced transplant-related mortality (TRM) and extended the upper age limit for allo-SCT [3,4,5]. However, the primary cause of treatment failure remains disease relapse, affecting approximately 40–50% of allo-SCT recipients [6,7], and the prognosis after disease relapse remains unfavorable [8,9,10].

The primary factors associated with a higher risk of post-transplant relapse encompass the absence of complete remission (CR) at the time of allo-SCT, high-risk cytogenetic characteristics, T-cell depletion, and the use of RIC regimens [11,12,13,14]. The current challenge lies in identifying the optimal strategy to manage the post-transplant phase in order to reduce disease recurrence risk or prevent it [15]. Interventions can be employed in three distinct settings: the maintenance phase, when there is no evidence of relapse; the pre-emptive phase in the presence of detectable minimal residual disease (MRD) or mixed chimerism to prevent overt relapse; or the therapeutic phase, when overt disease relapse has occurred [15,16,17]. Cellular immunotherapy, such as donor lymphocyte infusion (DLI), a second transplant, as well as the use of hypomethylating agents (HMAs) and innovative targeted therapies are the available therapeutic post-transplant options [18,19,20,21,22,23,24,25]. The selection of one treatment over another frequently relies on the patient’s clinical factors, including performance status and the presence of GVHD, as well as the type of relapse (early or overt). Due to the multitude of variables involved, a precise algorithm for the timing and selection of these treatments has not yet been defined [8].

In this scenario, we analyzed a cohort of 134 AML and MDS adult patients consecutively allotransplanted between 1 January 2015, and 31 December 2021 at Hospital Saint-Antoine AP-HP and Spedali Civili di Brescia and who subsequently relapsed. The aims of this study were to evaluate patient survival after allo-SCT relapse, to describe the type of relapse, and to identify the best therapeutic strategy to improve the outcome post-allo-SCT relapse. 

## 2. Materials and Methods

### 2.1. Study Design and Patients

From January 2015 to December 2021, a total of 553 AML/MDS patients were consecutively submitted for allo-SCT in the Bone Marrow Transplant Unit of Spedali Civili di Brescia (Italy) and Saint-Antoine University Hospital, Paris (France). Of these, 134 relapsed after allo-SCT (24.2%) and 103 (18.6%) were subsequently treated. Patients were categorized according to the type of relapse into two groups: the overt relapse (69/103 patients—67%) and the early relapse, which included patients with molecular/cytogenetic relapse and/or with mixed chimerism (34/103 patients—33%). Furthermore, within these two groups, we further divided patients based on whether they had received DLI (40/103—38.8%) or not (63/103—61.2%). All patients provided informed consent for data registration in the PROMISE database, in which clinical and biological data are collected. Supplementary data were extracted through a comprehensive review of patients’ clinical charts, encompassing both the transplant phase and the subsequent follow-up period. The study was conducted in compliance with current National and European legislation on clinical trials, in accordance with the Declaration of Helsinki and the principles of good clinical practice.

### 2.2. Definitions

Overt relapse was defined by the recurrence of blasts in peripheral blood (PB) or bone marrow (BM) infiltration by more than 5% of blasts. Early relapse was characterized by molecular or cytogenetic recurrence and/or mixed chimerism. Pre-emptive treatment was defined as therapy administration in cases of early relapse [26]. DLI was defined as the transfusion of unstimulated lymphocyte concentrates collected from the original stem cell donor as buffy coat preparations or as a transfusion of unmanipulated mobilized stem cells. 

At Spedali Civili di Brescia, the median dose of the first DLI was 1 × 10^6^ CD3+ cells/Kg, and in case of multiple infusions, an escalating schedule was chosen (5 × 10^6^ CD3+ cells/Kg for the second infusion, 10 × 10^6^ CD3+ cells/Kg for the third and 50 × 10^6^ CD3+ cells/Kg for the fourth). Lymphocyte doses in the case of haploidentical transplantation were reduced by 1 Log.

At Saint-Antoine University Hospital, the DLI doses were adapted depending on the setting (prophylaxis, pre-emptive, overt disease) and donor type (sibling, matched unrelated donor (MUD), mismatched unrelated donor (MMUD) or haploidentical). Briefly, focusing on the sibling and MUD 10/10 setting, the first dose of prophylactic DLI was 1 × 10^6^ CD3+/Kg, increasing by a half log each for the subsequent two doses. In the same setting of prophylaxis, considering MMUD and haploidentical donors, each dose level was reduced by a half log and by 1 Log, respectively. Moving from the prophylactic setting to the pre-emptive and overt disease setting, each dose level for each donor type was increased by a half log. 

The chimerism assessment on Italian patients in the Spedali Civili di Brescia laboratory was evaluated on BM CD34+cells by RT-qPCR (reverse transcription-quantitative polymerase chain reaction) short tandem repeat analysis [27,28]. The chimerism assessment on French patients in Saint-Antoine University Hospital was evaluated on PB CD33+ and CD3+ cells by RTqPCR [8,9]. Mixed chimerism was defined as failure to achieve >97.5% of donor cells, following data from our previous paper, which suggested this cut-off was able to significantly predict relapse [10]. Molecular monitoring of MRD was performed on common target genes (WT1, NPM1, FLT3-ITD) on PB or BM by RT-qPCR if these genes were detected at diagnosis. Complete remission (CR) was defined as the presence of <5% blasts in the BM and no circulating blasts in the PB [1]. Molecular complete remission (CRmol) was defined as MRD negativity [1]. Moreover, for analysis purposes, patients who re-obtained full donor chimerism were grouped together with those who obtained Crmol.

### 2.3. Statistical Analysis

Categorical data were presented as numbers and percentages, continuous data as median and range, respectively. The chi-squared test was used to test for differences among subgroups. The OS, defined as the interval from the date of post-allo-SCT relapse to death or last follow-up, was estimated using the Kaplan–Meier method [29]. The log-rank and Gray’s tests were employed to verify differences among the different groups. Univariate and multivariate analyses were performed using a Cox regression model [29]. In the univariate analysis, variables considered as possible prognostic factors were disease status at allo-SCT (CR vs. no CR), post-relapse therapy (yes vs. no), conditioning intensity (myeloablative vs. RIC), DLI administration (yes vs. no), acute GVHD (aGVHD) and chronic GVHD (cGVHD) after allo-SCT, donor type (sibling vs. MUD plus haploidentical donor), time of allo-SCT relapse > 1 year, and the type of relapse (overt vs. early). Multivariate analysis included all variables found to be significant at the level of *p* < 0.05. Statistical analyses were performed with EZR (version 4.2.2) [30].

## 3. Results

### 3.1. Patient and Transplant Characteristics

After allo-SCT relapse, 103 (77%) patients received therapy, and 31 (23%) did not due to their precarious clinical conditions. Table 1 shows the most important clinical and transplant characteristics of patients treated after allo-SCT relapse. The characteristics are listed according to the type of relapse, distinguishing between overt (n = 69) and early relapse (n = 34). 

In the overt relapse group, the median age at transplant was 60.1 years (20–74), and 39 patients (56.5%) were male. 84.1% of patients were diagnosed with AML, while 15.9% were diagnosed with MDS. 50.7% of patients received allo-SCT in CR, and 55.1% were in the first RC. PBSC was used in 94.2% of the cases, and the donor was sibling in 27.5% of the transplants, MUD/MMUD in 34.8% and haploidentical in 36.2%. The regime of conditioning was MAC in 53.6% of patients and RIC in the remaining 46.4%. GVHD prophylaxis consisted of cyclosporine (CsA) plus methotrexate (MTX) in 3 patients (4.3%), CsA plus MTX plus antithymocyte globulin (ATG) in 29 patients (21.7%), CsA plus mycofenolato mofetile (MMF) plus ATG in 16 patients (23.2%), post-transplant Cyclophosphamide PTCy based prophylaxis in 25 patients (36.2%), and other platforms in the remaining 10 patients. From the allo-SCT to the last follow-up, acute-GVHD (aGVHD) occurred in 34 (49.3) patients and chronic-GVHD (cGVHD) in 10 (14.5). The median time to relapse was 12 months (0.8–60.5).

In the early relapse group, the median age at transplant was 60.0 years (28.9–67.5), 21% of patients (61.8%) were male, and 31 (91.2%) were diagnosed with AML. 50% of patients received allo-SCT in CR, and 61.8% were in the first RC. PBSC was used in 91.2% of the cases, and the donor was sibling in 38.2% of the transplants, MUD/MMUD in 44.1% and haploidentical in 17.6%. The regime of conditioning was MAC in 55.9% of patients and RIC in the remaining 44.1%. GVHD prophylaxis consisted of CsA plus MTX in 3 patients (8.8%), CsA plus MTX plus ATG in 14 patients (41.2%), CsA plus MMF plus ATG in 8 patients (23.5%), post-transplant Cyclophosphamide PTCy based prophylaxis in 5 patients (14.7%), and other platforms in the remaining 4 patients. aGVHD was observed in 13 (38.2) patients and cGVHD in 11 (32.4) from the allo-SCT to the last follow-up. The median time to relapse was 12 months (1.0–56.3).

### 3.2. Post-Relapse Therapy and Response

Post-relapse therapy included DLI in 40 patients (38.8%), 20 of whom were in overt relapse and 20 were in early relapse. Among this group, 12 (30%) patients experienced aGVHD and 9 (22.5%) cGVHD, all of which resolved before DLI administration. Out of the 40 patients treated with a DLI-based regimen, 9 received DLI only, while the others also received HMA (4 cases), HMA + venetoclax (12 cases), FLT3-inhibitors (3 cases), intensive chemotherapy (5 cases), a second allo-SCT (5 cases), and other therapies (2 cases). 

Sixty-three patients (61.2%) were treated with a regimen, not including DLI, comprising 49 in overt relapse and 14 in early relapse. Among this group, 32 (50.8%) patients experienced aGVHD and of these, 8 had not resolved it at the time of relapse therapy, while 6 (9.5%) patients experienced cGVHD. Of these 63 patients, 10 received HMA, 27 HMA + venetoclax, 4 FLT3-inhibitors, 6 intensive chemotherapy, 9 a second allo-SCT, and 7 other therapies (Table 2).

Table 3 shows the response rate of patients treated after disease recurrence. Twenty-five of 40 (63%) patients treated with a DLI-based regimen achieved CRmol, and 15/40 (38%) patients did not respond. At the last follow-up, 24/40 (60%) patients treated with a DLI-based regimen had died, of which 18 (75%) due to disease relapse/progression and 6 (25%) due to other causes (3 for infections, 2 for GVHD, 1 during second allo-SCT). Nine out of 40 patients treated with DLI developed GVHD, comprising 3 cases of acute GVHD (1 grade I, 1 grade II, and 1 grade III) and 6 cases of chronic GVHD (2 mild, 2 moderate, and 2 severe). Among the two patients with severe chronic GVHD, both died of GVHD-related complications. The 1-, 2-, and 5-year cumulative incidence of GVHD (any grade and chronic) after DLI-based therapy was 10% (95% CI, 3–22%), 23% (95% CI, 11–39%) and 23% (95% CI, 11–39%), respectively (Figure 1).

A total of 16/63 (25%) patients treated without DLI achieved CRmol, and 47/63 (75%) did not respond. At the last follow-up, 52/63 (83%) patients who were treated without DLI had died, of which 48 (92%) due to disease relapse/progression and 4 (8%) due to other causes (2 for infections, 1 for GVHD, 1 for hepatic veno-occlusive disease) (Table 3). 

### 3.3. Survival Analysis after Post-Transplant Relapse

After a median follow-up of 1.6 years from relapse (range 0.21–8.06), the OS of this series of patients at 1, 2, and 5 years was 32% (95% CI 23–40%), 18% (95% CI 11–25%), and 11% (95% CI 7–20%), respectively (Figure 2a).

The 1-, 2-, and 5-year OS of patients treated after allo-SCT relapse was 40% (95% CI 29–49%), 20% (95% CI 14–31%), 15% (95% CI 7–24%), respectively, compared with 6% (95% CI 6–4%), 3% (95% CI 4–3%), 0% (95% CI 0–0%) for patients who did not receive therapy (*p* < 0.01) (Figure 2b).

Of the 103 treated patients, 34 (33%) received the treatment in early relapse, while 69 (67%) were treated in overt relapse. The OS at 1, 2, and 5 years was 60% (95% CI 45–70%), 36% (95% CI 19–54%), and 30% (95% CI 14–49%) for patients treated in early relapse vs. 26% (95% CI 16–38%), 12% (95% CI 5–22%), and 6% (95% CI 1–17%) for patients treated in overt relapse (*p* < 0.01) (Figure 3a).

Of the 103 patients who received therapy after relapse, patients who were treated with DLI-based regimens (n = 40) had an OS at 1, 2, and 5 years of 55% (95% CI 38–70%), 32% (95% CI 17–48%), and 32% (95% CI 17–48%), respectively. Patients treated with non-DLI regimens (n = 63) had an OS at 1, 2, and 5 years of 27% (95% CI 16–38%), 16% (95% CI 7–26%), 7% (95% CI 1–18%) (*p* < 0.01) (Figure 3b).

In the group of patients treated with DLI-based regimens, the 9 patients treated with DLI only showed an OS at 1, 2, and 5 years of 63% (95% CI 24–87%), 50% (95% CI 16–78%) and 50% (95% CI 16–78%), respectively, while the 31 patients treated with DLI and other therapies had an OS at 1, 2, and 5 years of 52% (95% CI 33–70%), 25% (95% CI 10–44%), and 19% (95% CI 6–38%) (*p* < 0.01) (Figure 4a).

We then compared the OS of patients treated with DLI-based regimens in early vs. overt relapse with the OS of patients treated without DLI in early vs. overt relapse. The OS of the 20 patients treated with DLI in early relapse at 1, 2, and 5 years was 67% (95% CI 42–84%),34% (95% CI 13–57%), and 34% (95% CI 13–57%), while the OS of the 20 patients treated with DLI in overt relapse at 1, 2, and 5 years was 43% (95% CI 20–65%), 20% (95% CI 4–44%), and 20% (95% CI 4–44%). The OS of the 14 patients treated without DLI in early relapse at 1, 2, and 5 years was 54% (95% CI 24–75%), 40% (95% CI 14–64%), and 26% (95% CI 5–54%), respectively, while the OS of the 49 patients treated without DLI in overt relapse at 1, 2, and 5 years was 17% (95% CI 8–29%), 5% (95% CI 1–16%), and 0% (95% CI 0–0%) (*p* < 0.01) (Figure 4b).

### 3.4. Univariate and Multivariate Analysis

Prognostic factors that were significantly (*p* < 0.05) associated with OS after allo-SCT relapse in the univariate proportional hazards model were related donor (HR 0.60, *p* < 0.01), cGVHD (HR 0.48, *p* < 0.01), time to relapse > 1 year (HR 0.62, *p* < 0.04), post-relapse treatment (HR 0.25, *p* < 0.01), overt relapse (HR 2.43, *p* < 0.01), DLI administration (HR 0.45, *p* < 0.01) (Figure 5).

The variables significantly associated with OS by univariate analysis were included in the multivariate analysis. As shown in Figure 6, time to relapse > 1 year (HR 0.59, *p* = 0.05), early relapse (HR 2.54, *p* < 0.01), and post-relapse treatment with DLI (HR 0.57, *p* = 0.03) were significantly associated with a better survival after allo-SCT relapse.

## 4. Discussion

Relapse following allo-SCT still represents the greatest obstacle against AML/MDS cure. As early as 2007, Schmid et al. [31] explored the role of DLI in the treatment of AML in overt hematological relapse. The OS at 2 years was 21% for patients receiving DLI and 9% for patients not receiving DLI. Among DLI recipients, factors such as lower tumor burden at relapse and remission at the time of DLI were predictive for survival in multivariate analysis. These findings prompted clinicians to identify the relapse as early as possible to adopt a pre-emptive treatment strategy [32]. The combination of MRD monitoring with lineage-specific molecular chimerism analysis appears to be the most sensitive way of detecting disease recurrence following allo-HCT [33]. Thus, in a recent work, the OS rate at 5 years after pre-emptive DLI for MRD/mixed chimerism was between 51% and 68% among responders and 37% among non-responders [34].

In this series of 134 AML/MDS patients relapsing after allo-SCT, the 2-year OS rate after allo-SCT relapse was estimated at 18%, which falls within the range reported in the existing literature (Figure 2a) [19,31,35]. In a recent European Society for Blood and Marrow Transplantation (EBMT) registry study on 8162 adult patients with AML who relapsed between 2000 and 2018 after allo-HCT, Bazabachi et al. [9] confirmed a dismal 2-year OS of around 17% for the entire cohort and observed a steady increase in 2-year survival from 16% to 26% among patients aged ≤ 50 years at relapse, likely reflecting, among other factors, the efficacy of post-transplant salvage including second allo-HCT. As expected, in our study, most (89%) deaths were attributed to the progression of the disease, highlighting the difficulty of relapse management. Nevertheless, treating patients with cellular therapy and/or innovative drugs after allo-SCT relapse results in better outcomes compared to palliative care, as shown in Figure 2b, and this prompts us to invest resources in identifying the best post-transplant salvage strategy.

To better understand how and when to act to improve survival after disease recurrence, we divided the patients who received therapy into two groups, based on type of relapse: overt and early, the latter including patients with recurrence of molecular/cytogenetic disease and/or with mixed chimerism (<97.5% donor cells). These two cohorts had almost completely overlapping characteristics. In the overt relapse group, haploidentical donors were more frequent, resulting in a higher number of patients who received PTCy as GVHD prophylaxis. While the difference in aGVHD was not statistically significant between the two groups, more patients treated in early relapse had cGVHD (*p* = 0.04) due to the higher number of patients receiving DLI in this context [34]. The better OS of patients treated in pre-emptive setting confirms the immunological efficacy of DLI in treating post-transplant leukemia recurrence, and also the central role of MRD monitoring after allo-SCT combining the early detection of molecular specific disease markers with post-transplant chimerism modifications.

The choice of therapy, including DLI or not, was based on the local policies adopted by the two centers during the last 6 years, largely influenced by the type of recurrence since DLIs are conventionally more frequently used when the recurrence is molecular or the chimerism is mixed [34,36] and in the absence of ongoing GVHD. Patients treated with DLI-based regimens had a significantly better OS compared to those treated with other therapies. The difference in OS between patients who received DLI only and patients treated with DLI in combination with other therapies reflects the early phase of relapse in the former group; notably, eight out of nine patients who received only DLI had early relapse.

When the OS is analyzed based on DLI administration (yes vs. no) and the phase of treatment administration (early vs. overt relapse), incorporating DLI into treatment regimens for overt relapse significantly improves the survival approach of patients treated in a pre-emptive setting. The use of new drugs such as epigenetic modulators, venetoclax, or FLT3 inhibitors likely enables a greater number of patients to reduce the leukemic burden and achieve remission, which can be further consolidated with subsequent DLI infusions. Furthermore, these data confirm the potentially curative effectiveness of graft-versus-leukemia even against advanced disease relapses [31,37,38,39]. As expected [40,41], the best outcomes were achieved when the relapse was promptly detected, and pre-emptive therapy was started, particularly if DLI was administered. In this latter case, it is worth noting that patients treated in early relapse with DLI reached a plateau in OS, which suggests that they may be cured.

Some limitations of our study must be considered. Firstly, patients were retrospectively evaluated; secondly, our results were derived from the analysis of subgroups that comprised relatively small numbers. This demands caution in drawing definitive conclusions and suggests that further studies on a larger number of patients are needed. Finally, MRD and chimerism were measured locally using different methods. Nevertheless, these results are in line with other data and suggest that even in the absence of randomized trials, the issue of MRD monitoring and pre-emptive treatment of relapse in the real world can be considered a mainstay of Good Clinical Practice.

In summary, our data confirm the poor outcome of AML/MDS patients who relapse after allo-SCT and show the real survival benefit obtained from a pre-emptive treatment strategy [26]. Which drugs or drug combinations are more effective has not yet been established and will probably be the focus of future studies. Moreover, the detection of MRD with highly specific and sensitive methods (e.g., digital PCR) [42,43] should be implemented in the future in order to progressively increase the number of patients who are treated pre-emptively with DLI.

## 5. Conclusions

The number of therapeutic opportunities to reduce the risk of disease relapse after allo-SCT for AML and MDS has increased in recent years, but the improvement in outcome following relapse after transplant is still an unmet clinical need. Integrating MRD detection after allo-SCT with chimerism monitoring and adopting an immunological pre-emptive intervention with DLI either alone or in combination might be a successful strategy [4]. The future priority must be the design of randomized trials, exploring the role of new targeted drugs and immunomodulating agents such as DLI in the different scenarios of the post-transplant follow-up, starting from the prophylaxis of disease relapse, and moving to the treatment of MRD (pre-emptive therapy) before overt relapse [26].

## Figures and Tables

**Figure 1 cancers-16-01278-f001:**
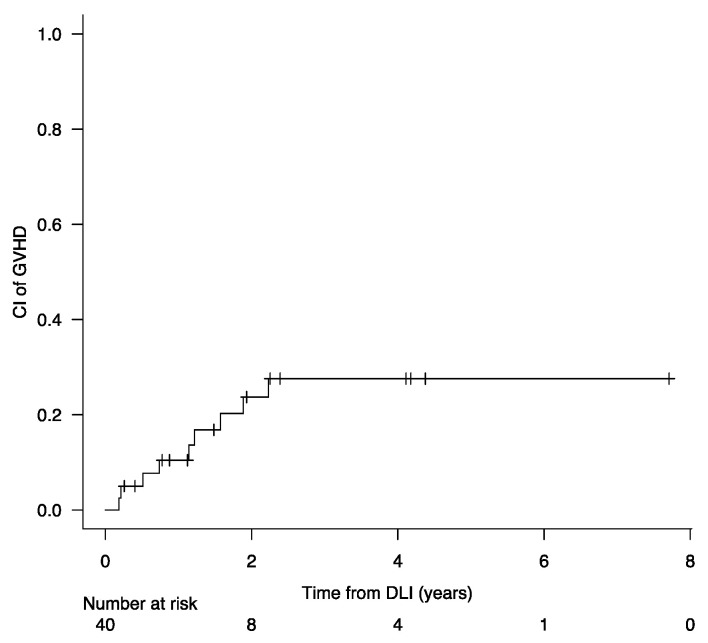
Cumulative incidence of any grade acute and chronic GVHD after DLI-based therapy.

**Figure 2 cancers-16-01278-f002:**
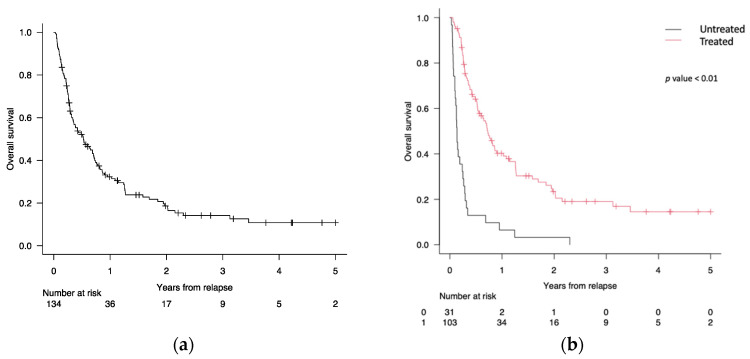
(**a**) Overall survival (OS) of relapsed patients after allo-SCT. (**b**) OS of 103/134 (77%) patients treated after allo-SCT relapse (red line) versus 31/134 (23%) patients untreated (black line).

**Figure 3 cancers-16-01278-f003:**
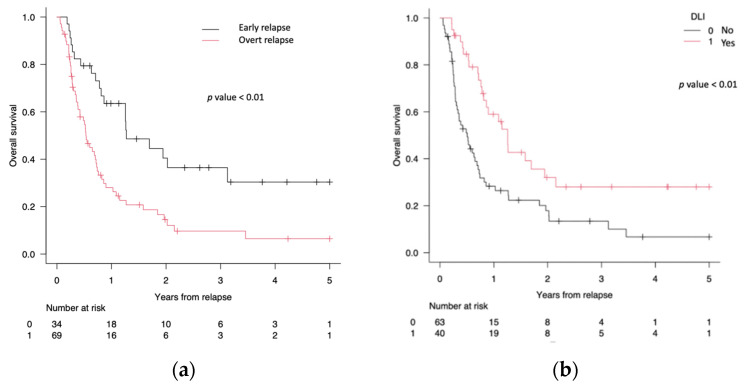
(**a**) Overall survival (OS) of 34/103 (33%) patients treated in early relapse (black line) versus OS of 69/103 (67%) patients treated in overt relapse (red line). (**b**) OS of 40/103 (39%) patients treated after allo-SCT relapse with a DLI-based regimen (red line) versus OS of 63/103 (61%) patients treated with a no-DLI-based regimen (black line).

**Figure 4 cancers-16-01278-f004:**
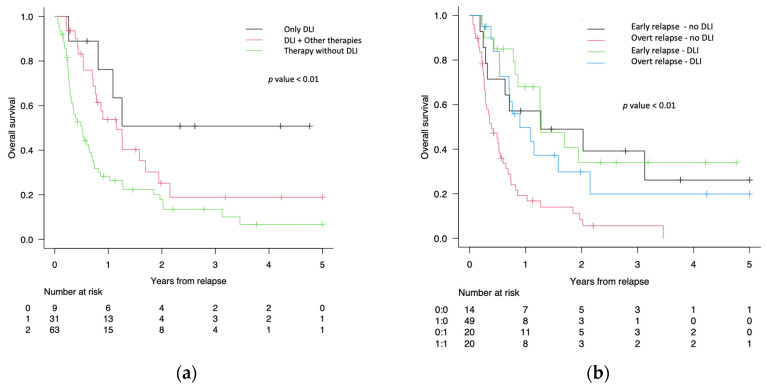
(**a**) Overall survival (OS) of the 9 patients treated after allo-SCT relapse with DLI only (black line) compared with OS of 31 patients treated with DLI plus other therapies (red line) and OS of 63 patients treated without DLI (green line). (**b**) OS of the patients treated with DLI-based regimens in early relapse (green line) versus overt relapse (blue line) versus OS of the patients treated without DLI in early relapse (black line) versus overt relapse (red line).

**Figure 5 cancers-16-01278-f005:**
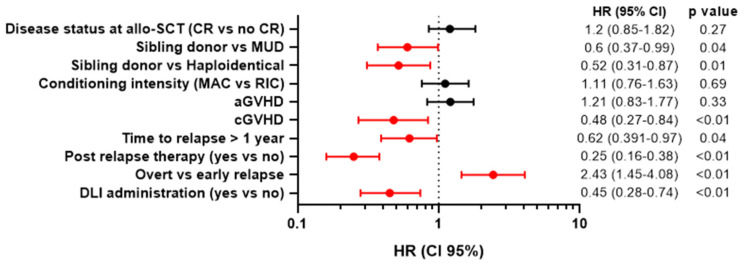
Univariate analysis of OS after allo-SCT relapse. allo-SCT—allogeneic stem cell transplant; MAC—myeloablative conditioning; RIC—reduced intensity conditioning; DLI—donor lymphocyte infusion; aGVHD—acute graft-versus-host disease; cGVHD—chronic graft-versus-host disease; MUD—matched unrelated donor; HR—hazard ratio.

**Figure 6 cancers-16-01278-f006:**
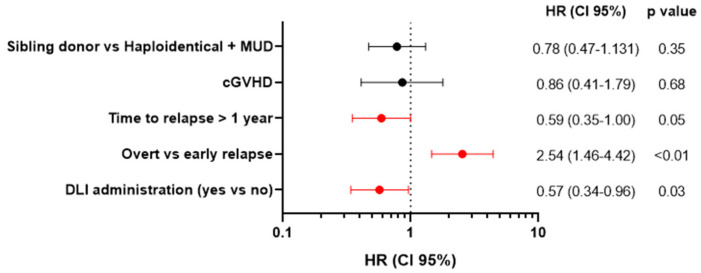
Multivariate analysis of OS after allo-SCT relapse. DLI—donor lymphocyte infusion; cGVHD—chronic graft-versus-host disease; MUD—matched unrelated donor; HR—hazard ratio.

**Table 1 cancers-16-01278-t001:** Clinical and transplant characteristics of the 103 patients treated after allogeneic stem cell transplant relapse.

	All Patients (*n* = 103)	Overt Relapse (*n* = 69)	Early Relapse (*n* = 34)	*p* Value
Age at allo-SCT, years, median (range)	60.1 (20–74)	60.1 (20–74)	60.0 (28.9–67.5)	1.00
Male, *n* (%)	60 (58.2)	39 (56.5)	21 (61.8)	0.67
Diagnosis, *n* (%)				0.38
AML	89 (86.4)	58 (84.1)	31 (91.2)	
MDS	14 (13.6)	11 (15.9)	3 (8.8)	
Disease status, *n* (%)				1.00
CR	52 (50.5)	35 (50.7)	17 (50)	
No-CR	51 (49.5)	34 (49.3)	17 (50)	
Lines of therapy, *n* (%)				0.53
1	59 (56.3)	38 (55.1)	21 (61.8)	
>1	44 (42.7)	31 (44.9)	13 (38.2)	
Donor type, *n* (%)				0.18
Sibling	32 (31)	19 (27.5)	13 (38.2)	
MUD/MMUD	39 (37.9)	24 (34.8)	15 (44.1)	
Haploidentical	31 (30)	25 (36.2)	6 (17.6)	
UCB	1 (1)	1 (1.4)	0 (0)	
Stem cell source, *n* (%)				0.60
PBSC	96 (93.2)	65 (94.2)	31 (91.2)	
BM	6 (5.8)	3 (4.3)	3 (8.8)	
UCB	1 (1)	1 (1.4)	0 (0)	
Conditioning regimen, *n* (%)				1.00
MAC	56 (54.4)	37 (53.6)	19 (55.9)	
RIC	47 (45.6)	32 (46.4)	15 (44.1)	
GVHD prophylaxis, *n* (%)				<0.01
CsA + MTX	6 (5.8)	3 (4.3)	3 (8.8)	
CsA + MTX + ATG	29 (28.2)	15 (21.7)	14 (41.2)	
CsA + MMF + ATG	24 (23.3)	16 (23.2)	8 (23.5)	
CsA/Sir + MMF + PTCy	7 (6.8)	3 (4.3)	4 (11.8)	
CsA + MMF + ATG + PTCy	23 (22.3)	22 (31.9)	1 (2.9)	
Other	14 (13.6)	10 (14.5)	4 (11.8)	
aGVHD	47 (45.6)	34 (49.3)	13 (38.2)	0.30
cGVHD	21 (20.4)	10 (14.5)	11 (32.4)	0.04
Time to relapse, months, median (range)	12 (0.8–60.5)	12 (0.8–60.5)	12 (1.0–56.3)	1.00
Follow-up, months, median (range)	1.65 (0.2–8.1)	1.65 (0.2–8.1)	1.61 (0.5–7.8)	1.00

Table legend: M—male; AML—acute myeloid leukemia; MDS—myelodysplastic syndrome; CR—complete remission; MUD—matched unrelated donor—MMUD—mismatched unrelated donor; PBSC—peripheral blood stem cells; BM—bone marrow; UCB—umbilical cord blood; MAC—myeloablative conditioning; RIC—reduced-intensity conditioning; CsA—cyclosporine A; Sir—sirolimus; MTX—methotrexate; ATG—antithymocyte globulin; PTCy—post-transplant cyclophosphamide; aGVHD—acute graft versus-host disease; cGVHD—chronic graft-versus-host disease; DLI—donor lymphocyte infusion.

**Table 2 cancers-16-01278-t002:** Post-relapse therapy of the 103 patients included in the study according to type of relapse.

	Overt Relapse, *n* (%)	Early Relapse, *n* (%)
Post-relapse therapy including DLI, *n* = 40	20 (50)	20 (50)
HMA	2 (10)	2 (10)
HMA + venetoclax	7 (35)	5 (25)
FLT3-inhibitors	2 (10)	1 (5)
Intensive chemotherapy	4 (20)	1 (5)
2nd allo-SCT	3 (15)	2 (10)
Other	1 (5)	1 (5)
Only DLI	1 (5)	8 (40)
Post-relapse therapy without DLI, *n* =63	49 (78)	14 (22)
HMA	6 (12)	4 (29)
HMA + venetoclax	24 (49)	3 (21)
FLT3-inhibitors	2 (4)	2 (14)
Intensive chemotherapy	5 (10)	1 (7)
2nd allo-SCT	7 (14)	2 (14)
Other	5 (10)	2 (14)

Table legend: DLI, donor lymphocyte infusion; HMA, hypomethylating agent; FLT3, FMS-related receptor tyrosine kinase 3; allo-SCT, allogeneic stem cell transplant.

**Table 3 cancers-16-01278-t003:** Response to therapy of 103 patients treated after allo-SCT relapse according to type of relapse.

	Overt Relapse, *n* (%)	Early Relapse, *n* (%)	*p* Value
Pts treated with DLI-based regimen, *n* = 40	20 (50)	20 (50)	
Response			
CRmol	14 (70)	11 (55)	
NR	6 (30)	9 (45)	0.51
GVHD post DLI	1 (5)	8 (40)	0.02
Deaths	13 (65)	11 (55)	0.75
Deaths due to disease progression	11 (55)	7 (35)	0.36
Pts treated without DLI, *n* = 63	49 (%)	14 (%)	
Response			
RCmol	8 (16)	8 (57)	
NR	41 (84)	6 (43)	<0.01
Deaths	43 (88)	9 (64)	0.06
Deaths due to disease progression	41 (84)	7 (50)	0.2

Legend: DLI—donor lymphocyte infusion; Pts—patients; CRmol—molecular complete remission; NR—no response; GVHD—graft-versus-host disease.

## Data Availability

Data are available by the corresponding author upon request.

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
