# Peer review of "Donor Lymphocyte Infusion in the Treatment of Post-Transplant Relapse of Acute Myeloid Leukemias and Myelodysplastic Syndromes Significantly Improves Overall Survival: A French–Italian Experience of 134 Patients"

_cancers, 2024, doi:10.3390/cancers16071278_

Round 1
Reviewer 1 Report
Comments and Suggestions for Authors
This study comprises a retrospective analysis of overall survival outcomes for patients treated with various modalities for recurrent AML and MDS. Several parameters were examined, including treatment with or without DLI, and time of administration of DLI (e.g., preemptive vs. treatment during overt recurrence). The authors report that significantly improved overall survival was achieved when DLI was administered in a preemptive manner. This fills a gap in the field, as currently there is no standardized treatment for salvage therapy after disease recurrence. The data are convincing, and appropriately interpreted. The manuscript is well-written with good discussions of the strengths and limitations of the study. Overall, this study has substantial merit in that it provides important new information regarding the best treatment approaches for patients who have had a recurrence after allo-SCT.
Author Response
Thank you very much for taking the time to review our manuscript.
Comments: This study comprises a retrospective analysis of overall survival outcomes for patients treated with various modalities for recurrent AML and MDS. Several parameters were examined, including treatment with or without DLI, and time of administration of DLI (e.g., preemptive vs. treatment during overt recurrence). The authors report that significantly improved overall survival was achieved when DLI was administered in a preemptive manner. This fills a gap in the field, as currently there is no standardized treatment for salvage therapy after disease recurrence. The data are convincing, and appropriately interpreted. The manuscript is well-written with good discussions of the strengths and limitations of the study. Overall, this study has substantial merit in that it provides important new information regarding the best treatment approaches for patients who have had a recurrence after allo-SCT.
Response: Thank you very much for appreciating our study.
Reviewer 2 Report
Comments and Suggestions for Authors
The authors report donor lymphocyte infusion in the treatment of post-transplant relapse of acute myeloid leukemias and myelodysplastic syndromes, significantly improves overall survival.
1. There is no data on chromosomal or genetic mutations in cases of leukemia and myelodysplastic syndrome. The authors should provide data and describe it in detail.
2. The numbers in the table are out of alignment. The authors should be corrected.
Author Response
Thank you very much for taking the time to review our manuscript. Please find the detailed responses below and the corresponding revisions/corrections in the re-submitted files.
Comments 1: There is no data on chromosomal or genetic mutations in cases of leukemia and myelodysplastic syndrome. The authors should provide data and describe it in detail.
Response 1: Thank you for this comment. We realize the importance of genetic mutations in case of leukemia and myelodysplastic syndrome and their role in the disease risk stratification. Nevertheless, this study has the limitations of a retrospective study. In particular, the data collected are those present in the PROMISE EBMT database where all transplanted patients are registered. Unfortunately, informations on the cytogenetics and molecular mutations of AML/MDS are often lacking in MED-A form, and the only way to obtain this information is to retrospectively review patients’ charts from 2015 to 2021 (553 records). Moreover, we believe that our numbers are too small to allow final conclusions on the role of cytogenetic and molecular data on the impact of the different strategies for post-transplant relapse
Comments 2: The numbers in the table are out of alignment. The authors should be corrected.
Response 2: Thank you for pointing this out. We revised the tables lay-out.
Reviewer 3 Report
Comments and Suggestions for Authors
The authors describe a medium-sized case series of patients with acute myeloid leukemia and MDS relapse after allogeneic stem cell transplantation. They describe the results for patients from the particular perspective of whether DLI was part of the treatment strategy and find better results after DLI. The results are generally of interest, but are not new overall. In recent 20 years, numerous papers have been published on the topic, which are easy to find even with a very cursory search (only including donor lymphocyte transfusion in the title of the papers) and are not sufficiently cited and discussed. The 2007 paper (ref 33) cited by the authors already makes a similar comparison to the one presented by them. Nevertheless, the results of this and other previous studies are not thoroughly discussed. The novelty value of the data presented is therefore low.
Within the data presentation is lacking:
The time interval from transplantation to recurrence
The extent of GVH in patients with or without DLI and the discussion of the interaction of these two factors.
Author Response
Thank you very much for taking the time to review our manuscript. Please find the detailed responses below and the corresponding revisions/corrections in the re-submitted files.
Comments 1:“In recent 20 years, numerous papers have been published on the topic, which are easy to find even with a very cursory search (only including donor lymphocyte transfusion in the title of the papers) and are not sufficiently cited and discussed. The 2007 paper (ref 33) cited by the authors already makes a similar comparison to the one presented by them. Nevertheless, the results of this and other previous studies are not thoroughly discussed”
Response 1:We have revised the initial section of the discussion (Line 296-304), presenting the findings from Schmid's 2007 paper published in JCO, alongside more recent results from BMT in 2022 (Schmid's, 2022) (Line 305-306).
Moreover, we chose the recently published EBMT Manuscript (Bazarbachi 2020) in order to enrich the discussion (Line 308-313)
Comments 2:Data lacking: The time interval from transplantation to recurrence.
Response 2:Thank you for pointing this out. We agree with this comment. We have added the time to relapse in Table 1 (pag 5) for both overt and early relapse patients.
Comments 3: Data lacking: The extent of GVH in patients with or without DLI and the discussion of the interaction of these two factors.
Response 3:Thank you for bringing this to our attention. The decision regarding therapy, whether to include DLI or not, was largely influenced by the nature of recurrence, as DLIs are conventionally utilized more frequently in cases of molecular recurrence or mixed chimerism and by the absence of ongoing GVHD. For patients received DLO, we have added the type of GVHD (acute or chronic GHVD) and the GVHD grading (line 211-2014) and we highlighted that two patients with severe GHVD died of GVHD-related complications.
Round 2
Reviewer 2 Report
Comments and Suggestions for Authors
none
Author Response
Thank you very much for taking the time to review our manuscript.
Reviewer 3 Report
Comments and Suggestions for Authors
Literature on the subject is now cited in more detail.
The question about the frequency of GVHD was of course not only referring to GVHD after DLI, but in particular to the question of how often GVHD had occurred in the patients before relapse and whether GVH was present at the time of the relapse. As the authors themselves point out, this is an essential parameter for the decision as to whether DLIs will be given. In this respect, it is imperative to provide this information separately for patients treated with versus without DLIs.
Author Response
Thank you very much for taking the time to review this manuscript. Please find the detailed responses below and the corresponding revisions/corrections in the re-submitted files.
Comments 2 :The question about the frequency of GVHD was of course not only referring to GVHD after DLI, but in particular to the question of how often GVHD had occurred in the patients before relapse and whether GVH was present at the time of the relapse. As the authors themselves point out, this is an essential parameter for the decision as to whether DLIs will be given. In this respect, it is imperative to provide this information separately for patients treated with versus without DLIs.
Response 2:Thank you for pointing this out, we are agree with this comment. We have reviewed the frequency of GVHD in patients who underwent treatment with DLI and those who did not. Among the patients treated with DLI, 12 experienced aGVHD and 9 experienced cGVHD, all of which resolved prior to DLI administration. Among the patients treated without DLI, 32 experienced aGVHD and of these 8 had not resolved it at the time of therapy and 6 patients experienced cGVHD. We added this information to lines 193-194 and 199-200.
Round 3
Reviewer 3 Report
Comments and Suggestions for Authors
Information about GvHD before or at relapse now provided. Information on resolution of cGvHD missing in part. Influence of GvHD on treatment decision at relapse and resulting data analysis should be discussed.